# Lift Yourself Up: Retrieval-augmented Text Generation with Self-Memory

**Xin Cheng**[1]   **Di Luo**[2]   **Xiuying Chen**[3]   **Lemao Liu**[4]   **Dongyan Zhao**[1]   **Rui Yan**[2]

[1] Peking University   [2] Remin University of China
[3] KAUST   [4] Tencent AI Lab
chengxin1998@stu.pku.edu.cn

## Abstract

With direct access to human-written reference as memory, retrieval-augmented generation has achieved much progress in a wide range of text generation tasks. Since better memory would typically prompt better generation (we define this as *primal problem*). The traditional approach for memory retrieval involves selecting memory that exhibits the highest similarity to the input. However, this method is constrained by the quality of the fixed corpus from which memory is retrieved. In this paper, by exploring the duality of the primal problem: better generation also prompts better memory, we propose a novel framework, `Selfmem`, which addresses this limitation by iteratively employing a retrieval-augmented generator to create an unbounded memory pool and using a memory selector to choose one output as memory for the subsequent generation round. This enables the model to leverage its own output, referred to as self-memory, for improved generation. We evaluate the effectiveness of `Selfmem` on three distinct text generation tasks: neural machine translation, abstractive text summarization, and dialogue generation, under two generation paradigms: fine-tuned small model and few-shot LLM. Our approach achieves state-of-the-art results in four directions in `JRC-Acquis` translation dataset, 50.3 ROUGE-1 in `XSum`, and 62.9 ROUGE-1 in `BigPatent`, demonstrating the potential of self-memory in enhancing retrieval-augmented generation models. Furthermore, we conduct thorough analyses of each component in the `Selfmem` framework to identify current system bottlenecks and provide insights for future research[1].

## 1   Introduction

In recent years, retrieval-augmented text generation has attracted growing interest across various fields, including neural machine translation[28, 17, 2], dialogue response generation[81, 6, 46], and language modeling[36, 77, 19]. This innovative generation paradigm initially equips a fine-tuned small model or a large language model (LLM) with access to an external database (typically the training corpus) using information retrieval techniques. Subsequently, the generation process is conducted based on both the input text and the retrieved memory.

In this paradigm, the guiding principle for memory retrieval is to find the memory that exhibits the highest similarity to the current input [36, 96, 49]. This aligns with the human intuition that a more similar demonstration sample typically offers more hints. As demonstrated in Figure 1, for a retrieval-augmented translation model, the memory similarity alone exhibits a strong correlation with the final translation quality, regardless of other factors that may influence translation quality (e.g.,

---

[1]Code and data available at: `https://github.com/Hannibal046/SelfMemory`

37th Conference on Neural Information Processing Systems (NeurIPS 2023).

polysemy, morphology, and coreference). We define this as the *primal problem*: **better memory prompts better generation**. Consequently, numerous studies have focused on how to retrieve better memory, ranging from sparse retrieval to dense retrieval [10, 63], from a fixed retriever to a learnable retriever [41, 8], and from sentence-level memory to more fine-grained token-level memory [36, 35].

However, a fundamental limitation exists in all previous works: the memory is retrieved from a fixed corpus and is constrained by the corpus's quality. Due to the finite retrieval space, bounded memory significantly restricts the potential of memory-augmented generation models [97]. In this paper, we explore the duality of the *primal problem*, which posits that **better generation also prompts better memory**. We propose a novel framework called `Selfmem`, which iteratively employs a retrieval-augmented generator to create an unbounded memory pool and uses a memory selector to choose one output as memory for the subsequent generation round. By combining the *primal* and *dual problem*, a retrieval-augmented generation model can elevate itself using its own output, referred to as self-memory. The key insight behind `Selfmem` is that the text more closely resembling the data distribution during inference is not the training data [87], but the model's own output.

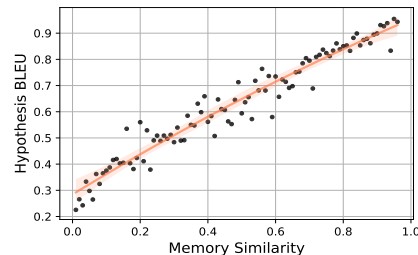

Figure 1: Relation between memory and hypothesis on `JRC-Acquis` En→De dataset. The hypothesis is generated by a retrieval-augmented translator whose memory is retrieved from the training set. The X-axis represents the similarity between memory and the reference.

`Selfmem` consists of two complementary components: a retrieval-augmented generator and a memory selector. The generator operates under two distinct paradigms: fine-tuning a small model or few-shot prompting an LLM. For the former, we train the generator with labeled data and retrieved memory, while for the latter, we employ a fixed black-box LLM exclusively for inference alongside retrieved in-context learning samples. We then use the generator's output to train a memory selector based on a specific performance metric. By simply replacing the retrieved memory with unbounded generated memory, we achieve higher-quality generation output (*primal problem*), which subsequently serves as memory for the next round after being refined by the memory selector (*dual problem*).

To evaluate the efficacy of the `Selfmem`, we carry out comprehensive experiments in three distinct text generation tasks: neural machine translation, abstractive text summarization, and dialogue generation. We witness substantial enhancements over robust baselines, attaining state-of-the-art outcomes in `JRC-Acquis` (four directions), `XSum` (50.3 ROUGE-1), and `BigPatent` (62.9 ROUGE-1). To gain deeper insights into the `Selfmem`, we meticulously investigate each crucial component and pinpoint the existing system bottleneck to guide future research endeavors.

## 2 Related Work

### 2.1 Retrieval-augmented Text Generation

Since the world is not a snapshot once the training corpus is collected, we can never expect an ever-large model to capture everything in its parameters, even for LLMs like GPT-4 [62]. Therefore, it is crucial to equip these models with an external memory bank to store additional knowledge or useful demonstration examples for solving various NLP tasks[41, 78, 95].

In the translation domain, retrieval techniques have long been employed by the localization industry to enhance human translators' productivity and consistency even before the advent of machine translation [94]. Early works on machine translation primarily focused on utilizing memory for statistical machine translation (SMT) systems [80, 50]. For neural machine translation (NMT), [28] were the first to use search engines to retrieve memory from the training set and incorporate it with an external memory network. Subsequent research explored various aspects of retrieval-augmented NMT, such as memory encoding methods [92, 93, 31], joint training of retrievers and generators with monolingual data [8], memory granularity [35], and memory diversity [17]. For few-shot LLM generation, strategies for in-context example selection have been proposed to improve translation quality [2]. Furthermore, in-context machine translation has been shown to be effective for on-the-fly adaptation [79]. For dialogue response generation tasks, employing exemplar/template

retrieval as an intermediate step has proven advantageous for generating informative responses [89, 91, 6, 7]. In-context learning example retrieval also aids in controllable dialogue [46]. Other applications include abstractive summarization [64, 14, 18, 15], code generation [30], paraphrase generation [34, 83], language modeling [36, 105], counterfactual data generation [24], open domain question answering [12, 33] and semantic parsing [99].

## 2.2 Neural Text Reranking

By alleviating the discrepancy between training and inference (i.e., exposure bias) and directly optimizing desired metrics, two-stage reranking methods have facilitated significant progress in various text generation tasks. In machine translation, pioneering works by [75] and [61] introduced and popularized discriminative reranking for SMT. In the context of NMT, research has focused on two primary reranking approaches: generative reranking [56, 32, 88] and discriminative reranking [39, 71, 23]. For syntactic parsing, [21] were the first to employ a two-stage reranking method to select outputs from a base parser, while [11] introduced a maximum entropy reranker. In text summarization, RefSum [53] proposed a second-stage summarization framework to address train-test distribution mismatches. SimCLS [54] used pairwise Learning To Rank (LTR) to select candidates with the highest matching scores. SummaReranker [68] adopted a multi-task mixture-of-experts framework to leverage different metrics capturing various aspects of generated candidates. BRIO [55] reused the base model for a second round of fine-tuning with both cross-entropy loss and a candidate-level ranking loss. JGR [76] employed an alternate training paradigm to train the generator and reranker.

A key limitation of these reranking methods is that they only represent a one-way process, wherein the selected candidates become the system's final output. In contrast, our framework innovatively utilizes the chosen candidates as memory for the subsequent generation round of a retrieval-augmented generator, which can produce better candidates with enhanced memory.

## 3 Methods

In this section, we begin with a motivating experiment on *generation as memory* (§ 3.1). Then, we introduce Selfmem, a framework comprising a *retrieval-augmented generator* (§ 3.2) and a *memory selector* (§ 3.3). The complete framework and algorithm are illustrated in Figure 2 and Algorithm 1.

## 3.1 Generation as Memory

The primary motivation behind our framework stems from the observation that the memory, which is more similar in distribution to the data during inference, is not the training data (38.89 BLEU, as shown in the first row of Table 1). Instead, it is the model's own output (58.58 BLEU) within the unbounded generation space. One interesting exploration involves directly utilizing the generated output as memory in relation to the *primal problem*: better memory prompts better generation.

We conduct experiments on the JRC-Acquis En→De dataset. The first row in Table 1 represents conventional retrieval-augmented training with retrieved memory and achieves a 58.58 BLEU score. However, directly incorporating beam output of this trained model as memory (Beam) back into the generation model does not yield any improvements (row 2), despite its higher similarity to the reference compared to the retrieved ones. We hypothesize two potential reasons for this: (1) the retrieval-augmented generator may not generalize effectively in this context due to the

Table 1: Experiments on the relation between memory quality and the final hypothesis quality, measured by the BLEU score with ground truth translation. The retrieval-augmented translator keeps fixed while the memory is obtained from different sources.

| Memory Source | Memory Quality | Hypothesis Quality |
|---------------|----------------|--------------------|
| Retrieval | 38.89 | 58.58 |
| Beam | 58.58 | 58.43 |
| Reference | 100 | 90.43 |
| Random | 1.14 | 49.08 |

memory distribution shift (from 38.89 to 58.58), and (2) the beam memory does not offer any information gain compared to the retrieved one, even it exhibits more overlap with the references.

To investigate the first hypothesis, we conduct experiments under the oracle and random scenarios by using the reference as memory (Reference) and randomly sampled sentences as memory (Random). The result is shown in Table 1 and it illustrates that a retrieval-augmented generator (trained with

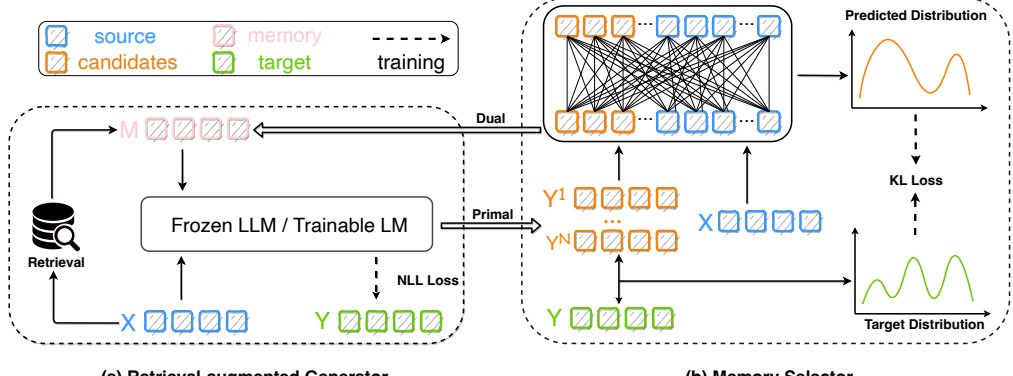

**Figure 2:** Overall framework. There are two components in `Selfmem`, a retrieval-augmented generator (a) and a memory selector (b). For the primal problem, (a) takes source and memory as input to generate candidates for (b). For the dual problem, (b) takes as input source and generated candidates to select memory for (a).

retrieved memory) has already learned to discriminate between different memories in both oracle and random scenarios, without updating the model weights.

To evaluate the second conjecture, we first define the token sets of the reference, retrieved memory, and beam memory as $\mathcal{R}, \mathcal{M}$, and $\mathcal{B}$, respectively. The overlap token set, denoted by $\mathcal{O}$, is defined as the tokens that overlap with the references in the beam memory but not in the retrieved memory, which is represented as $\mathcal{R} \cap \mathcal{B} - \mathcal{R} \cap \mathcal{M}$. $\mathcal{O}$ is considered as the additional information provided by the beam memory. Inspired by the confidence analysis of NMT model [58], we compute the set confidence score, $\psi(\cdot)$, as follows:

$$\psi(\cdot) = \frac{1}{|\cdot|} \sum_{y^i \in \cdot} p(y_i | x, y_{<i}) \tag{1}$$

where $p(y_i | x, y_{<i})$ is defined by the generation model. $\psi(\cdot)$ measures the confidence with which the generation model generates the tokens. The value of $\psi(\mathcal{R})$ is 0.58, while that of $\mathcal{O}$ is 0.76, indicating that the generator is relatively confident in generating tokens in $\mathcal{O}$, and therefore does not need to resort to external memory [38]. Beam search ranks generated candidates based on $p(y|x)$, where the selected memory falls within the confidence region of the generator and consequently provides no information gain. This observation motivates us to select memory according to metrics other than $p(y|x)$ in the memory selector (§3.3).

### 3.2 Retrieval-augmented Generator

Given a text pair $(x, y)$, where $x = \{\mathrm{x}_1, ..., \mathrm{x}_{|x|}\}$ is the source, $y = \{\mathrm{y}_1, ..., \mathrm{y}_{|y|}\}$ is the target. They could be (document, summary) in summarization, (context, response) in dialogue generation or (source, target) in machine translation. The retrieval-augmented generation would first use $x$ to retrieve memory $m$ from datastore $\mathbb{D}$. Then the generator $G_\xi(x, m)$, parameterized by $\xi$, would take both $x$ and $m$ as input to generate the target sentence $y$. In this paper, following standard practice, we choose the training set as $\mathbb{D} = \{(x^i, y^i)\}_{i=1}^{|\mathbb{D}|}$. For LLM as $G_\xi$, we use the standard in-context learning format to give $(x, y)$ as demonstration example. For tunable generator $G_\xi$, we only keep the target side of *top-1* retrieval results as memory and we consider two commonly used architectures: **Joint-Encoder** [29, 87, 41] and **Dual-Encoder** [92, 8, 17].

**Joint-Encoder** This architecture is the standard encoder-decoder-based model [3, 84]. The input is the concatenation of $x$ and $m$. The encoder would first map the input into the hidden states $H$:

$$H = \mathrm{Encoder}(x \text{ [SEP] } m) \tag{2}$$

And the decoder would incorporate $H$ by attention mechanism and generate tokens in an auto-regressive manner:

$$h^i = \text{Decoder}(\text{CrossAttn}(H), y_{<i}) \quad P_{G_\xi}(\cdot|x, y_{<i}) = \text{Softmax}(h^i) \tag{3}$$

**Dual-Encoder** Instead of treating $x$ and $m$ as a long sequence, this architecture has two encoders, one for $x$ and the other for $m$. Their outputs are sequentially attended by the decoder with dual cross attention as in [17]:

$$H_x = \text{SourceEncoder}(x) \quad H_m = \text{MemoryEncoder}(m) \tag{4}$$

$$h^i = \text{Decoder}(\text{CrossAttn}(H_x, H_m), y_{<i}) \tag{5}$$

We use Transformer [84] as the building block for both architectures and optimize $G_\xi$ with NLL loss:

$$\mathcal{L}_{\text{nll}} = -\sum_{t=1}^{|y|} \log P_{G_\xi}(y_t|x, m, y_{<t}) \tag{6}$$

### 3.3 Memory Selector

The role of memory selector $S_\theta(x, c)$, parameterized by $\theta$, is to select one candidate $c$ from the candidate pool $\mathbb{C}$ generated by $G_\xi$ based on a specific metric $\Delta(\cdot, \cdot)$. The chosen candidate $c$ is then utilized as memory $m$ for the subsequent generation round of $G_\xi$. As discussed in §3.1, using $p_{G_\xi}(y|x)$ as the metric $\Delta(\cdot, \cdot)$ would result in falling into the confidence region of $G_\xi$, leading to no information gain. Moreover, a larger value of $p_{G_\xi}(y|x)$ does not necessarily guarantee improved generation quality [59]. Consequently, we define $\Delta(\cdot, \cdot)$ as model-free metrics that are widely employed for assessing generation quality, such as BLEU for Neural Machine Translation (NMT) and ROUGE for Summarization. Our memory selector takes the concatenation of the source $x$ and candidate $c_i$ as input, and produces a multinomial distribution $p_{S_\theta}(\cdot|x)$ over $\mathbb{C}$.

In this paper, we focus on the role of the memory selector, $S_\theta(x, c)$, which is parameterized by $\theta$. The objective of this selector is to choose a single candidate $c$ from the candidate pool $\mathbb{C}$, generated by $G_\xi$, based on a specific metric, $\Delta(\cdot, \cdot)$.

$$p_{S_\theta}(c_i|x) = \frac{\exp(S_\theta(x \,[\text{SEP}]\, c_i))}{\sum_{j=1}^{|\mathbb{C}|} \exp(S_\theta(x \,[\text{SEP}]\, c_j))} \tag{7}$$

In accordance with [39], the training goal for $S_\theta$ is to minimize the discrepancy between the $S_\theta$'s predictions and the scores determined by $\Delta(\cdot, \cdot)$. This divergence is quantified using the Kullback-Leibler (KL) divergence.

$$\mathcal{L}_{\text{kl}} = -\sum_{i=1}^{|\mathbb{C}|} p_M(c_i) \log p_{S_\theta}(c_i|x) \quad \text{where} \quad p_M(c_i) = \frac{\exp(\Delta(c_i, y)/\tau)}{\sum_{j=1}^{|\mathbb{C}|} \exp(\Delta(c_j, y)/\tau)} \tag{8}$$

$\tau$ is the temperature to control the smoothness of the distribution. At inference, the output of the $S_\theta$ is $\arg\max\limits_{c_i \in \mathbb{C}} p_{S_\theta}(c_i|x)$.

### 3.4 Combine Generator and Selector

We define two generation modes for $G_\xi$. The first mode, referred to as the *hypothesis mode*, generates a single output for each input, which is utilized for system evaluation. The second mode, known as the *candidate mode*, produces N outputs for a given input, and is employed for training $S_\theta$ as well as memory selection. By integrating two modes together, we present the complete framework of our proposed model, `Selfmem`, as illustrated in Algorithm 1.

## 4 Experimental Setup

### 4.1 Dataset

We assess the performance of `Selfmem` on three generation tasks, utilizing a total of seven datasets. **Translation.** We evaluate our framework on `JRC-Acquis` datasets [82], a collection of parallel

**Algorithm 1** `Selfmem` Framework

---

**Require:** a dataset $\mathbb{D}$, a retriever $R$, a memory selection metric $\Delta(\cdot, \cdot)$, a retrieval-augmented generator $G_\xi$, and a memory selector $S_\theta$
1: retrieve memory $\mathbb{M}$ in $\mathbb{D}$ with $R$
2: train $G_\xi$ with $\mathbb{D}$ and $\mathbb{M}$ (if not LLM)
3: use $G_\xi$ to generate candidate pool $\mathbb{C}$ with $\mathbb{M}$ in candidate mode
4: train $S_\theta$ on $\mathbb{C}$ with $\Delta(\cdot, \cdot)$
5: **while** not converged in the validation set **do**
6:     $S_\theta$ selects memory from $\mathbb{C}$ as $\mathbb{M}$
7:     $G_\xi$ generates candidate pool $\mathbb{C}$ with $\mathbb{M}$ in candidate mode
8: **end while**
9: $G_\xi$ generates the final hypothesis with $\mathbb{M}$ in hypothesis mode

---

legislative text of European Union Law. It is the benchmark dataset used in translation memory-augmented NMT task [28, 92, 8, 17]. We choose 4 translation directions, namely, Spanish↔English (Es↔En), German↔English (De↔En). **Summarization.** We evaluate on 2 summarization datasets: 1) `XSum` [60], extreme summarization, a single-document summarization dataset with highly abstractive articles from British Broadcasting Corporation. 2) `BigPatent` [73], consisting of 1.3 million records of U.S. patent documents along with human-written abstractive summaries. **Dialogue.** We experiment on `DailyDialog` [44], which contains multi-turn dialogs on daily life topics and is used by [13, 4, 103]. The detailed statistics for these datasets can be found in the Appendix A.

## 4.2 Implementation Details

We utilize the BM25 algorithm [70] for retrieval purposes. For all tasks, the candidate generation method consists of beam search with a beam width of 50. The number of iterations is determined by the performance on the validation set. **For translation**, we follow the approach of [93, 8, 17], employing a randomly initialized Transformer$_{\text{base}}$ architecture as $G_\xi$ for trainable small model and XGLM [48] for LLM in-context learning. Evaluation metrics include BLEU, TER, and chrF++ obtained from SACREBLEU[66]. The memory selector $S_\theta$ utilizes an XLM-R$_{base}$[22] as backbone, with BLEU serving as $\Delta(\cdot, \cdot)$. **For summarization**, we initialize $G_\xi$ with BART$_{\text{base}}$[40] for `BigPatent` and employ BRIO [55] for `XSum`. The evaluation metric comprises ROUGE (R-1/2/L) [47]. **For dialogue generation**, BART$_{\text{base}}$ serves as the backbone for $G_\xi$. Our dialogue system is evaluated using BLEU (B-1/2) and Distinct (D-1/2) scores [43]. For both dialogue and summarization tasks, we adhere to the methods of [54, 26], adopting RoBERTa$_{\text{base}}$ [52] as the backbone for $S_\theta$. The linear combination of B-1/2 is chosen as $\Delta(\cdot, \cdot)$ for Dialogue Generation, while R-1/2/L is used for Summarization, following [76]. For further implementation details, please refer to the Appendix B and Appendix C for evaluation metrics.

# 5 Experimental Results

## 5.1 Machine Translation

We select four translation directions and experiment with two generation paradigms: trainable small models and few-shot prompted LLMs [85, 20]. For trainable models, we explore two architectures (joint and dual, as detailed in §3.2). The baselines comprise two types of translation systems: one being the vanilla sequence-to-sequence model [3, 84] without memory augmentation, and the other consisting of retrieval-augmented translation models focusing on memory encoding [28, 92], memory construction [101], memory retrieval [8], and memory diversity [17]. Based on the experimental results[2] shown in Table 2, `Selfmem` significantly enhances the performance of $G_\xi$ across four translation datasets and two different architectures. This is noteworthy, given that the parameters of the $G_\xi$ remain fixed, with the only variable being the input memory. This finding is consistent with the *primal problem* which posits that improved memory typically leads to better generation results.

---

[2]As higher BLEU scores in this range do not necessarily guarantee a superior translation system [9], we also evaluate our system using TER and chrF++. The results can be found in the Appendix D.

Table 2: Results of translation task on `JRC-Acquis` measured by BLEU. Models denoted by the same symbol (⋆ and †) have the same parameters and only differ in memory as input. The bolded numbers show the SOTA performance and the underlined numbers show the second-best result. ∗ denotes the system is significantly better than baselines with *p-value* < 0.05 tested by [37].

| System | Es→En | | En→Es | | De→En | | En→De | |
|---|---|---|---|---|---|---|---|---|
| | Dev | Test | Dev | Test | Dev | Test | Dev | Test |
| **None Memory** | | | | | | | | |
| RNNsearch [3] | 55.02 | 59.34 | 50.54 | 50.48 | 50.20 | 49.74 | 44.94 | 43.98 |
| Transformer [84] | 64.08 | 64.63 | 62.02 | 61.80 | 60.18 | 60.16 | 54.65 | 55.43 |
| **Retrieval Memory** | | | | | | | | |
| SEG-NMT [28] | 60.28 | 59.34 | 57.62 | 57.27 | 55.63 | 55.33 | 49.26 | 48.80 |
| NMT-pieces [101] | 63.97 | 64.30 | 61.50 | 61.56 | 60.10 | 60.26 | 55.54 | 55.14 |
| G-TFM [92] | 66.37 | 66.21 | 62.50 | 62.76 | 61.85 | 61.72 | 57.43 | 56.88 |
| MonoNMT [8] | 67.73 | 67.42 | 64.18 | 63.86 | 64.48 | 64.62 | 58.77 | 58.42 |
| CMM [17] | 67.48 | 67.76 | 63.84 | 64.04 | 64.22 | 64.33 | 58.94 | 58.69 |
| Transformer$_{dual}$⋆ | 66.87 | 67.12 | 63.14 | 63.54 | 64.09 | 63.36 | 58.69 | 58.06 |
| Transformer$_{uni}$† | 67.74 | 67.32 | 63.93 | 64.12 | 64.50 | 64.40 | 58.16 | 58.58 |
| **Self-Memory** | | | | | | | | |
| Transformer$_{dual}$⋆ | **68.63**∗ | **69.20**∗ | 64.12∗ | 64.67∗ | 65.06∗ | 64.98∗ | 59.26∗ | 59.49∗ |
| Transformer$_{uni}$† | 68.26∗ | 68.80∗ | **66.07**∗ | **65.94**∗ | **65.32**∗ | **65.65**∗ | **59.88**∗ | **60.11**∗ |

Table 3: Comparison between retrieval memory and self-memory. The quality of memory and hypothesis is measured by the n-gram overlap with reference (BLEU). All experiments are conducted with Transformer$_{joint}$ on `JRC-Acquis`.

| | | Retrieval | | Self | |
|---|---|---|---|---|---|
| | | memory | hypothesis | memory | hypothesis |
| **En-De** | → | 38.89 | 58.58 | 57.92 | 60.11 |
| | ← | 42.56 | 64.40 | 64.32 | 65.65 |
| **En-Es** | → | 40.67 | 64.12 | 63.57 | 65.94 |
| | ← | 43.05 | 67.32 | 67.78 | 68.80 |

The *dual problem* is revealed in Table 3. Self-memory, which essentially represents the model's own output, exhibits greater similarity with the ground truth and serves as a more effective memory for generating the final output. This observation highlights a key distinction between `Selfmem` and previous reranking works [39, 68]. Reranking aims to select candidates of higher quality than the beam output, whereas in `Selfmem`, the chosen candidates serve as memory for the retrieval-augmented generator and do not necessarily need to surpass the quality of the beam hypotheses.

Table 4: Evaluation results of in-context learning with self-memory.

| | | XGLM-1.7B | | | XGLM-4.5B | | | XGLM-7.5B | | |
|---|---|---|---|---|---|---|---|---|---|---|
| | | Random | kNN | Self | Random | kNN | Self | Random | kNN | Self |
| En-De | → | 11.51 | 37.87 | 40.94 | 17.51 | 37.60 | 38.25 | 18.48 | 47.82 | 48.32 |
| | ← | 27.42 | 51.00 | 51.88 | 30.62 | 48.12 | 48.36 | 33.03 | 55.65 | 55.12 |
| En-Es | → | 23.87 | 46.20 | 48.56 | 31.83 | 48.37 | 49.17 | 29.97 | 53.86 | 54.32 |
| | ← | 25.29 | 51.55 | 53.13 | 32.16 | 48.55 | 49.22 | 35.22 | 57.25 | 57.56 |

In Table 4, we present the results of LLM with self-memory. We employ XGLM [48] as our backbone generator, with three different sizes ranging from 1.7B to 7.5B. We utilize the recommended prompt as described in [48]. We select three in-context learning examples and report the average scores from three separate runs, taking into account the sensitivity of example selection in ICL [49]. From the table, we first observe a general trend where few-shot translation performance improves as the

size of the model increases. Furthermore, we find that more similar translation demonstrations significantly enhance performance across all model sizes (from random, kNN to Self). This suggests that demonstration examples in in-context learning not only act as triggers for model ability but also adhere to the *primal problem*, where better demonstration example leads to better generation. Also, by comparing the results in Table 2 and Table 4, we can conclude that the cross-lingual LLM with designed examples still falls short of the supervised baselines in this task.

## 5.2 Summarization

In this paper, we compare the performance of our trainable model with those of REINA [87], PEGASUS [100], and BART [40]. The results are presented in Table5. Initially, it can be observed that memory has varying impacts on different datasets. The enhancement brought by memory in the `BigPatent` dataset is significantly larger than that in the `XSum` dataset. This can be attributed to the inherent characteristics of the `BigPatent` dataset, which consists of official patent documents that exhibit considerable similarity. Consequently, this greatly improves the summarization quality in accordance with the *primal problem*. Furthermore, we discovered that self-memory substantially enhances the performance of both BRIO (+1.2 R1) and BART (+18.5 R1), achieving state-of-the-art results on both datasets. We selected these baselines for a fair comparison, as they share the same base generator. Due to space constraints, additional comparisons and the confidence region of the SOTA model can be found in the Appendix E.

Table 5: Results of summarization task on `XSum` and `BigPatent` measured by ROUGE.

| System | Memory | R-1 | R-2 | R-L | | System | Memory | R-1 | R-2 | R-L |
|---|---|---|---|---|---|---|---|---|---|---|
| **XSum** | | | | | | **BigPatent** | | | | |
| PEGASUS | None | 47.2 | 24.6 | 39.3 | | PEGASUS | None | 53.6 | 33.2 | 43.2 |
| BRIO | None | 49.1 | 25.6 | 40.4 | | BART | None | 44.4 | 21.3 | 31.0 |
| REINA (PG) | Retrieval | 48.2 | 26.0 | 40.2 | | REINA (B) | Retrieval | 59.5 | 42.6 | 50.6 |
| REINA (B) | Retrieval | 43.2 | 21.0 | 35.5 | | REINA (L) | Retrieval | 60.7 | 43.3 | 51.3 |
| REINA (L) | Retrieval | 46.5 | 24.1 | 38.6 | | REINA (PG) | Retrieval | 44.6 | 21.5 | 33.3 |
| BRIO$_{dual}\star$ | Retrieval | 48.6 | 26.1 | 40.6 | | BART$_{dual}\star$ | Retrieval | 57.4 | 43.3 | 49.7 |
| BRIO$_{joint}\dagger$ | Retrieval | 49.5 | 26.5 | 41.2 | | BART$_{joint}\dagger$ | Retrieval | 59.6 | 43.4 | 51.0 |
| BRIO$_{dual}\star$ | Self | 49.2 | 26.2 | 40.8 | | BART$_{dual}\star$ | Self | 61.2 | 44.6 | 52.3 |
| BRIO$_{joint}\dagger$ | Self | **50.3** | **26.7** | **41.6** | | BART$_{joint}\dagger$ | Self | **62.9** | **48.1** | **59.6** |

## 5.3 Dialogue Generation

As demonstrated in Table 6, the self-memory significantly enhances the performance of the retrieval-augmented generator for dialogue generation tasks. By optimizing memory using BLEU as $\Delta(\cdot, \cdot)$, the self-memory improves the B-1,2 score over retrieved memory by 3.08 B-1 and 0.6 B-2 on BART$_{joint}$. Intriguingly, although `Selfmem` surpasses the baselines in terms of B-1/2, it falls behind in D-1 and D-2, which can be attributed to the trade-off between BLEU score and Distinct score when evaluating a dialogue system [104]. To address this issue, we opt for D-1,2 as $\Delta(\cdot, \cdot)$ when optimizing $S_\theta$, denoted as BART$_{joint}\dagger$(D). The results in Table 6 highlight the remarkable flexibility of `Selfmem` by directly optimizing memory to achieve the desired attributes for diverse and informative dialogue.

## 6 Further Analysis

To gain a deeper insight into `Selfmem`, we first examine the impact of each key component, namely $G_\xi$ and $S_\theta$. Subsequently, we perform a detailed token-level analysis of the generated output concerning their frequency in the training set. Experiments are conducted on the `JRC-Acquis` En→De dataset. We also include latency analysis and human evaluation on Appendix F and G.

**Tuning $S_\theta$** We explored various $S_\theta$ by direct selection from the candidate pool based on gold rankings. As shown in Figure 3a, both architectures with enhanced $S_\theta$ significantly outperform the current SOTA performance (60.11 BLEU). Moreover, we assessed the candidate pool quality during this iterative process using an oracle $S_\theta$, as displayed in Figure 3b. A clear pattern emerges

Table 6: Results of dialogue generation task on `DailyDialog` measured by B-1/2 and D-1/2. $BART_{joint}$ (D) denotes the metric $\Delta(\cdot, \cdot)$ for $S_\theta$ is the average of D-1 and D-2.

| System | Memory | B-1 | B-2 | D-1 | D-2 |
|---|---|---|---|---|---|
| NCM [86] | None | 33.60 | 26.80 | 3.00 | 12.80 |
| iVAE [25] | None | 30.90 | 24.90 | 2.90 | 25.00 |
| PLATO-2 [5] | None | 34.80 | 25.12 | 3.54 | 25.11 |
| DialoFlow [45] | None | 36.17 | 27.67 | 4.56 | 27.12 |
| BART | None | 20.72 | 11.36 | 3.92 | 19.44 |
| $BART_{dual}\star$ | Retrieval | 29.50 | 21.89 | 4.74 | 26.01 |
| $BART_{joint}$† | Retrieval | 36.72 | 31.55 | 6.13 | 35.65 |
| $BART_{dual}\star$ | Self | 33.43 | 22.85 | 4.66 | 26.16 |
| $BART_{joint}$† | Self | **39.80** | **32.15** | 5.84 | 32.16 |
| $BART_{joint}$† (D) | Self | 36.92 | 32.09 | **9.12** | **37.05** |

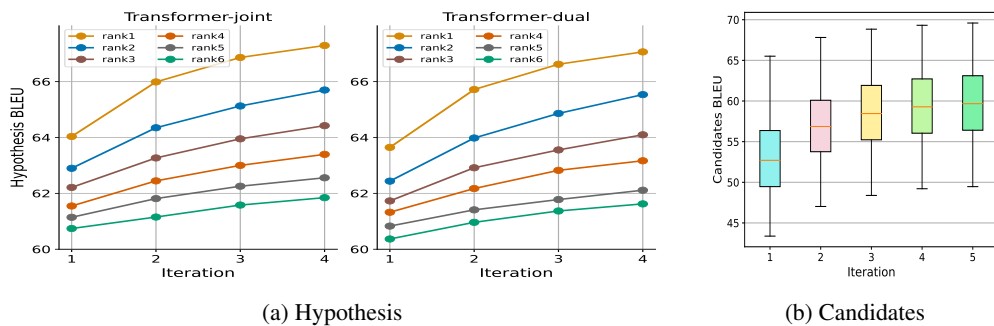

(a) Hypothesis  (b) Candidates

Figure 3: (a) shows generation quality in the iteration process with different $S_\theta$ in both trainable generator architectures. (b) shows candidates quality in the iteration process with an oracle $S_\theta$.

in this boxplot, revealing improvements in the *oracle*, *quartile*, *average*, and *minimum* scores of the candidate pool. These two experiments jointly clarify the `Selfmem`'s underlying intuition: a retrieval-augmented generator profits from superior memory, which can be chosen from its own unbounded output, and subsequently, the generator with improved memory produces a higher-quality candidate pool for the next selection round. Consequently, the model lift itself up.

**Tuning $G_\xi$**   As discussed in §3.1, we demonstrated that a trained retrieval-augmented generator, with fixed parameters, possesses the ability to distinguish between "good" and "bad" memory. This observation not only justifies our decision to maintain a fixed generator within our framework but also implies that the $G_\xi$ is not the current bottleneck of the `Selfmem`.

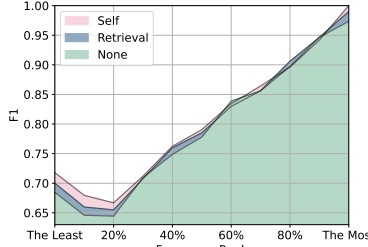

Figure 4: 1-gram F1 score sorted by training corpus frequency.

**Frequency Analysis**   We conduct a comprehensive token-level analysis by computing the 1-gram F1 scores for generated translations and subsequently categorizing the tokens based on their frequency in the training set. The results are depicted in Figure 4. A noticeable pattern emerges, suggesting that the more frequently a model encounters a token during training, the higher the accuracy of the generated output [102]. Moreover, our findings indicate that retrieval-augmented models, particularly those incorporating self-memory augmentation, exhibit superior performance in handling long-tail inputs which are challenges for parametric models [67, 57].

# 7 Conclusion

For the first time, we investigate the fundamental limitation of bounded memory in the current retrieval-augmented literature. We combine the *primal* and *dual problems* together and propose `Selfmem`, a general framework for retrieval-augmented text generation by uplifting generation model with its own output. We conduct comprehensive experiments across various text generation tasks and different generation paradigms, including trainable small model and few-shot prompted LLM. We surpass strong baselines and improve the state-of-the-art performance in serval datasets. We also meticulously investigate each crucial component and pinpoint the existing system bottleneck to guide future research endeavors.

## Limitations

We discuss the limitations of our framework as follows:

(1) Although `Selfmem` greatly improves the generation quality compared with other retrieval-augmented generation models, it requires more computational resources with respect to the memory selection process. For large dataset with long context (e.g., BigPatent), it would become a more crucial problem considering the quadratic time complexity of transformer architecture.

(2) This paper proposes a general idea for the retrieval-augmented generation. But we only experiment with transformer-based architecture for both generator and memory selector and the architecture of generator and memory selector keeps the same across all text generation tasks. We believe the task-specific design for the model architecture, training objective and generation methods in different text generation scenarios would further improve the performance.

## Acknowledgement

This work was supported by the National Key Research and Development Program of China (No.2021YFC3340304) and National Natural Science Foundation of China (NSFC Grant No.62122089). We appreciate the anonymous reviewers for their helpful comments. Dongyan Zhao and Rui Yan are the corresponding authors.

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

# A  Dataset Details

Table 7: Dataset statistics for three tasks.

| Task | Dataset | #Train | #Dev | #Test |
|---|---|---|---|---|
| Translation | JRC (en ↔ de) | 663,487 | 2,454 | 2,483 |
| | JRC (en ↔ es) | 653,127 | 2,533 | 2,596 |
| Summarization | BigPatent | 1,207,222 | 67,068 | 67,072 |
| | XSum | 204,045 | 11,332 | 11,334 |
| Dialogue | DailyDialog | 87,170 | 8,069 | 7,740 |

# B  Self Memory Details

For machine translation tasks, following [93, 8, 17] we use randomly initialize Transformer$_{base}$ architecture [84] as $G_\xi$. We use the joint-bpe algorithm [72] and share the parameters between the memory encoder and source encoder for dual encoder architecture. The hyper-parameter setting follows [17] with dropout 0.1, label smoothing 0.1, gradient clipping 1.0, Adafactor [74], warm-up steps 4000, maximum learning rate 4.4e-2 and training epochs 30 for total. The evaluation metrics are BLEU, TER and chrF++ from SACREBLEU [66]. The backbone of memory selector $S_\theta$ is XLM-R$_{base}$ [22] with BLEU as $\Delta(\cdot, \cdot)$. The hyper-parameter setting for $S_\theta$ follows [39] with $\tau$ 0.5, minmax normalization for candidates ranking, Adam optimizer with max learning rate 5e-5 and polynomial decay scheduler, and classifier dropout 0.2.

For Summarization, we init the $G_\xi$ with BART$_{base}$ [40] for BigPatent following [87] and state-of-the-art BRIO [55] for XSum. Optimization is based on Adafactor with a maximum learning rate of 5e-3, warm-up steps 10000 and gradient clipping value 1.0. The maximum input length is 512 for XSum and 1024 for BigPatent. The evaluation metric is Rouge (R-1/2/L) [47].

For Dialogue Generation, we use BART$_{base}$ as the backbone for $G_\xi$ on DailyDialog. We tune the hyper-parameters from learning rate {5e-3,1e-3,4e-4} and set dropout 0.1, batch size 64, label smoothing factor 0.1, maximum input length 120 for DailyDialog. Following [4, 13], we evaluate our dialogue system with BLEU (B-1/2) and Distinct (D-1,2) [43]. For both Summarization and Dialogue Generation task, we follow [54, 26] and adopt RoBERTa$_{base}$ [52] as the backbone for $S_\theta$. We choose the linear combination of B-1/2 as $\Delta(\cdot, \cdot)$ for Dialogue Generation and R-1/2/L for Summarization following [76]. We tune the hyper-parameters $\tau$ from {0.08,0.2,0.5,0.8}, learning rate from {5e-5,7e-5,2e-4}. The maximum input length for $S_\theta$ is 512 and we truncate tokens from the longer input of source and candidate.

# C  Evaluation Details

**Machine Translation**  We evaluate our MT system with BLEU, TER and chrF++ from SACRE-BLEU[3] [66]. The signatures for BLEU, TER and chrF++ are shown in Table 8.

Table 8: Signature from SACREBLEU.

| [c]**Signature** |
|---|
| nrefs:1\|case:mixed\|eff:no\|tok:13a\|smooth:exp\|version:2.0.0 |
| nrefs:1\|case:lc\|tok:tercom\|norm:no\|punct:yes\|asian:no\|version:2.0.0 |
| nrefs:1\|case:mixed\|eff:yes\|nc:6\|nw:2\|space:no\|version:2.0.0 |

---

[3]`https://github.com/mjpost/sacrebleu.git`

**Summarization** We evaluate our Summarization system with standard ROUGE [47] Perl package[4] for evaluation. Following [55], we use PTB tokenizer[5] for tokenization. And the parameters for ROUGE are "-c 95 -r 1000 -n 2 -m".

**Dialogue Generation** Following [27], we evaluate our dialogue system with NLTK BLEU [6] with space as tokenizer and smoothing method1. The Distinction score is from [42].

## D   More results on translation tasks

Table 9: Evaluation results on `JRC-Acquis` En→De measured by BLEU, TER and chrF++.

| System | Memory | BLEU ↑ | chrF++ ↑ | TER↓ |
|---|---|---|---|---|
| Transformer | None | 55.43 | 70.31 | 36.35 |
| Transformer$_{dual}$ | Retrieval | 58.06 | 71.58 | 35.41 |
| Transformer$_{joint}$ | Retrieval | 58.58 | 72.22 | 34.39 |
| Transformer$_{dual}$ | Self | 59.49 | 72.62 | 34.04 |
| Transformer$_{joint}$ | Self | **60.11** | **73.25** | **32.62** |

## E   More Summarization Baselines

In this Table 10, we include more baselines on the benchmark dataset `XSum` and `BigPatent`. We also report the confidence region of SOTA model for XSum and BigPatent as shown in Table 11.

Table 10: More baselines on `XSum` and `BigPatent`.

| System | R-1 | R-2 | R-L |
|---|---|---|---|
| XSum | | | |
| [51] | 38.8 | 16.5 | 31.3 |
| [40] | 45.1 | 22.3 | 37.3 |
| [100] | 47.2 | 24.6 | 39.3 |
| [54] | 47.6 | 24.6 | 39.4 |
| [55] | 49.1 | 25.6 | 40.4 |
| [87](PG) | 48.2 | 26.0 | 40.2 |
| [87](B) | 43.1 | 21.0 | 35.5 |
| [87](L) | 46.5 | 24.1 | 38.6 |
| [68] | 48.1 | 25.0 | 40.0 |
| [69] | 47.1 | 24.1 | 38.8 |
| [16] | 47.8 | 25.0 | 39.7 |
| Selfmem | **50.3** | **26.7** | **41.6** |

| System | R-1 | R-2 | R-L |
|---|---|---|---|
| BigPatent | | | |
| [100] | 53.6 | 33.1 | 42.3 |
| [40] | 44.4 | 21.3 | 31.0 |
| [98] | 60.6 | 42.5 | 50.0 |
| [65] | 38.7 | 12.3 | 34.1 |
| [90] | 45.0 | 20.3 | 39.2 |
| [1] | 52.3 | 33.5 | 42.8 |
| [87] (B) | 59.5 | 42.6 | 50.6 |
| [87] (L) | 60.7 | 43.3 | 51.3 |
| [87] (PG) | 44.6 | 21.5 | 33.3 |
| Selfmem | **62.9** | **48.1** | **59.6** |

## F   Empirical analysis of latency

In Table 12, we present empirical results of `Selfmem` latency, measured in seconds. We compare `Selfmem` with a retrieval-augmented baseline model across various datasets and computational platforms, including CPU and CUDA. The number of iterations for `Selfmem` is set to one. All experiments are conducted on the same device, equipped with one NVIDIA A100 GPU and one AMD EPYC 7V13 64-Core Processor.

---

[4]`https://github.com/summanlp/evaluation/tree/master/ROUGE-RELEASE-1.5.5`
[5]`https://nlp.stanford.edu/nlp/javadoc/javanlp/edu/stanford/nlp/process/`
`PTBTokenizer.html`
[6]`https://www.nltk.org/_modules/nltk/translate/bleu_score.html`

Table 11: Confidence region for SOTA model in XSum and BigPatent.

| System | ROUGE-1/2/L | 95%-conf.int |
|--------|-------------|--------------|
| | XSum | |
| BRIO$_{joint}$ | 50.3 | 0.49986 - 0.50602 |
| | 26.7 | 0.26300 - 0.26989 |
| | 41.6 | 0.41231 - 0.41900 |
| | BigPatent | |
| BART$_{joint}$ | 62.9 | 0.62664 - 0.63080 |
| | 48.1 | 0.47783 - 0.48333 |
| | 59.6 | 0.59401 - 0.59847 |

Table 12: Generation Latency analysis.

| | | NMT | XSum | BigPatent | DailyDialog |
|--|--|-----|------|-----------|-------------|
| Average Input Length | | 87 | 512 | 1024 | 71 |
| Average Output Length | | 44 | 75 | 127 | 16 |
| **CPU** | | | | | |
| Retrieval-augmented Baseline | | 0.97 | 1.79 | 3.16 | 0.32 |
| Selfmem | Candidate Generation | 3.20 | 7.50 | 15.00 | 1.02 |
| | Memory Selection | 0.50 | 0.52 | 0.95 | 0.14 |
| | Hypothesis Generation | 0.97 | 1.79 | 3.00 | 0.32 |
| | | ×4.80 | ×5.47 | ×6.04 | ×4.63 |
| **CUDA** | | | | | |
| Retrieval-augmented Baseline | | 0.29 | 0.44 | 0.75 | 0.10 |
| Selfmem | Candidate Generation | 0.51 | 1.00 | 1.72 | 0.18 |
| | Memory Selection | 0.01 | 0.01 | 0.01 | 0.01 |
| | Hypothesis Generation | 0.29 | 0.44 | 0.75 | 0.10 |
| | | ×2.76 | ×2.99 | ×3.35 | ×2.91 |

# G   Human and GPT-4 Evaluation

We employ both human annotators and GPT-4 (gpt-4-0314) annotators to perform pairwise ranking of the output generated by `Selfmem` and baseline systems. For GPT-4 annotators, we utilize the prompt from Alpaca Eval [7]. We randomly select 50 samples for translation tasks and 20 samples for summarization and dialogue tasks. The win rate of `Selfmem` versus retrieval-augmented baselines is depicted in Figure 1.

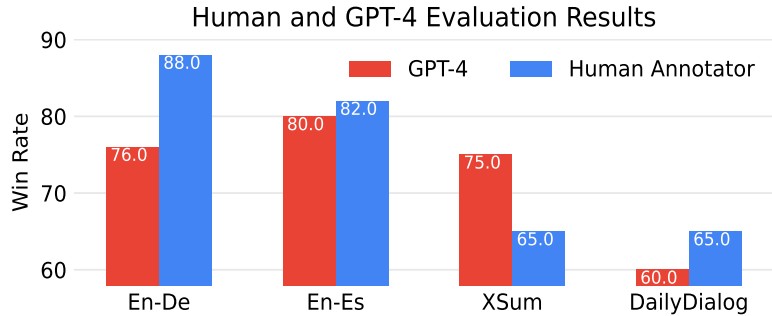

Figure 5: Human and GPT-4 evaluation results.

---

[7]`https://github.com/tatsu-lab/alpaca_eval/blob/main/src/alpaca_eval/evaluators_`
`configs/alpaca_eval_gpt4/alpaca_eval.txt`

