# OpenReview forum: "Lift Yourself Up: Retrieval-augmented Text Generation with Self-Memory"
_NeurIPS.cc/2023/Conference — NeurIPS 2023 poster_

### Official Review · Reviewer_Ysdi · 2023-07-05

**Soundness:** 3 good
**Presentation:** 3 good
**Contribution:** 3 good
**Rating:** 6
**Confidence:** 3

**Summary:**

The paper proposes a novel framework called Selfmem to address the limitations of retrieval-augmented text generation. Compared with memory retrieval from a fixed corpus, Selfmem iteratively uses a retrieval-augmented generator to create an unbounded memory pool and select the candidate output as memory for the subsequent generation round. The paper evaluates the effectiveness of Selfmem on three text generation tasks and achieves state-of-the-art results. The paper also conducts thorough analyses of each component in the framework.

**Strengths:**

The motivation of the primal problem and its duality is fairly intriguing. Preliminary experiments and supporting evidence demonstrate the rationality of the proposed method.

The proposed Selfmem achieves promising results on machine translation, summarization and dialogue generation, indicating its practical applicability.

The paper is overall clearly written. The problem setting and proposed approach is lucidly presented.


**Weaknesses:**

There is a growing body of recent work [1] that is increasingly using LLMs to generate knowledge instead of retrieving it from external corpus. This paper shares similar ideas with the new paradigm and should incorporate and discuss it in the related work section.

Further clarification and empirical evaluation on the diversity of retrieval memory from Selfmem should be provided. Given the limited knowledge encoded in the trainable generator, it remains uncertain whether Selfmem can effectively handle knowledge-intensive tasks compared to retrieval from external sources.

Selfmem appears to require more computational cost than direct retrieval. However, there is no further discussion on time computation in the paper.

[1] Yu et al. Generate rather than Retrieve: Large Language Models are Strong Context Generators. ICLR 2023


**Questions:**

1.	Although you mention the computational resources in the limitations section, can you offer an empirical comparison between Selfmem with other retrieval-based methods?

2.	For a more vivid demonstration, can you provide any case studies or examples to illustrate how the generated contexts from Selfmem outperform the retrieval-based ones.


**Limitations:**

See the weakness section.

---

> ### Author Rebuttal · Authors · 2023-08-07
>
> Firstly, we would like to express our sincere gratitude for the time and effort you have dedicated to reviewing our paper. We will address each comment in a point-by-point manner.
>
> - Comment 1: discussion about using LLM to generate knowledge instead of retrieval
> - Response 1: Thank you for mentioning the relevant reference by Yu et al. and saving us from LLM paper flood. We would include this paper, as well as any other pertinent studies, in the final version of our work.
>
> ---
>
> - Comment 2: Evaluation of knowledge-intensive tasks
> - Response 2: We appreciate your suggestion to incorporate knowledge into our framework for evaluation on knowledge-intensive tasks. However, it is important to note that the primary focus of this paper is not on this aspect. In the retrieval-augmented literature, there are generally two types of working systems: one that directly retrieves kNN samples from training set, wherein task knowledge is implicitly expressed through format, style, and word selection for tasks such as machine translation[1] and language modeling[2]; and the other that focuses on external knowledge retrieval for knowledge-intensive tasks, such as open domain QA[3] and fact verification[4]. Our work is primarily concerned with the former type and we leave the latter for future exploration.
>
> [1] Urvashi Khandelwal et al. Nearest Neighbor Machine Translation
>
> [2] Urvashi Khandelwal et al. Generalization through Memorization: Nearest Neighbor Language Models
>
> [3] Gautier Izacard et al. Leveraging Passage Retrieval with Generative Models for Open Domain Question Answering
>
> [4] Patrick Lewis et al. Retrieval-Augmented Generation for Knowledge-Intensive NLP Tasks
>
> ---
>
> - Comment 3: Empirical results of computation overhead
> - Response 3: Thank you for highlighting this important aspect. We have provided a detailed analysis of the computation overhead of Selfmem in comparison to baseline systems in the global rebuttal pdf file. We hope this addresses your concerns.
>
> ---
>
> - Comment 4: Request for a more vivid demonstration
> - Response 4: Upon the paper's acceptance, we will release all outputs, including those from Selfmem and baselines, and we will make an effort to identify examples that best showcase our method.

---

> > ### Comment · Reviewer_Ysdi · 2023-08-19
> >
> > Thank you for the detailed response with commitments. I think the overall quality of this paper is good. I will keep my score as "Weak Accept".

---

> ### Author Response · Authors · 2023-08-18
>
> Dear Reviewer, I hope this message finds you well. As the discussion period for our paper  is nearing its end, we kindly request your response to our rebuttal letter. We value your insights and are eager to address any concerns you may have. Your timely feedback would be greatly appreciated and will help us improve our paper. Thank you for your time and dedication to the review process.

---

### Official Review · Reviewer_Mggy · 2023-07-05

**Soundness:** 3 good
**Presentation:** 3 good
**Contribution:** 2 fair
**Rating:** 5
**Confidence:** 4

**Summary:**

The paper presents a framework, Selfmem, aimed at enhancing text generation tasks via memory retrieval. The core uniqueness of the framework resides in its capacity to create an unbounded memory pool by utilizing its retrieval-augmented generator and memory selector components iteratively. This allows the model to leverage its output, referred to as self-memory, for improved generation. The authors apply this framework into neural machine translation, abstractive text summarization, and dialogue generation tasks, demonstrating its efficacy with state-of-the-art results

**Strengths:**

(1) The paper proposes the Selfmem framework, which addresses the constraint of fixed corpus in memory retrieval.
(2) The paper demonstrates the effectiveness of Selfmem on various text generation tasks.
(3) The paper provides detailed analyses of components within the Selfmem framework, revealing potential bottlenecks and providing  insights.


**Weaknesses:**

(1) It would be better if more experiments beyond neural machine translation, abstractive text summarization, and dialogue generation tasks could be conducted. It will be beneficial to see its application in other related areas since the proposed method is task-agnostic.
(2) Much of the paper's experimentation relies on certain metrics (ROUGE-1) for evaluation. The universality of the findings might be cross-checked through the usage of other performance metrics.
(3) Though the authors mention learnable retrievers, they don't discuss how the Selfmem framework may be adapted for use with such retrievers.


**Questions:**

(1) Have you considered using other evaluation methods (e.g., human evaluation) to further validate the performance of the framework?
(2) How can the Selfmem framework be adapted for optimal usage with learnable retrievers?


**Limitations:**

A potential concern is, compared to human-written texts, machine-generated texts are noisier and with potential bias. Hence, is the proposed method able to avoid being affected by its own bias?

---

> ### Author Rebuttal · Authors · 2023-08-07
>
> Firstly, we would like to express our sincere gratitude for the time and effort you have dedicated to reviewing our paper. We will address each comment in a point-by-point manner.
>
> - Comment 1: More generation tasks needed.
> - Response 1:  Thank you for recognizing the versatility of our method. In our current study, we have chosen three generation tasks across seven datasets to assess our framework. This decision is based on two main factors: (1) we believe that these three tasks, specifically machine translation, summarization, and dialogue generation, effectively represent text generation tasks, and (2) these tasks have been extensively employed in retrieval-augmented literature [1][2][3], providing a relatively comparable testbed for benchmarking our method. Additionally, we plan to make our code open-source, allowing researchers to verify its effectiveness in other tasks.
>
> [1] Jason Weston et al. Retrieve and Refine: Improved Sequence Generation Models for Dialogue
>
> [2] Jiatao Gu et al. Search Engine Guided Neural Machine Translation
>
> [3] Nabil Hossain et al. Simple and Effective Retrieve-Edit-Rerank Text Generation
>
> ---
>
> - Comment 2: Metrics other than ROUGE for evaluation
> - Response 2: First, we would like to clarify that we did not heavily rely on ROUGE-1 for evaluation. For various generation tasks with distinct desired attributes, we employed different well-acknowledged metrics for assessment. Moreover, to cross-check our findings, we have included additional evaluation results of TER and chrF++ for translation tasks in the appendix. Furthermore, human evaluation and GPT-4 evaluation results are available in the global response PDF file for your reference.
>
> ---
>
> - Comment 3: Combination with learnable retriever
> - Response 3:  Although we briefly mentioned learnable retrievers in our paper, we do not believe it is of significant relevance to Selfmem. As illustrated in Algorithm 1, the retriever's role within our framework is mainly confined to the initial step. While a learnable retriever might outperform a non-learnable one, incorporating it would not fundamentally change our framework. This is because the core concept of Selfmem lies in generating memory rather than retrieving it, based on the observation that the quality of retrieved memory is always constrained by the memory pool. Furthermore, the empirical results in Table 2 provide a direct comparison between our framework and a generation model with a learnable retriever (MonoNMT), demonstrating the superiority of self-memory over retrieved-memory.
>
> ---
>
> - Comment 4: Debias from machine-generated text
> - Response 4: Thank you for bringing up this intriguing issue. We believe that our framework has the potential to reduce bias in machine-generated text. Considering that the entire framework is optimized by the memory selection metrics $\Delta(\cdot,\cdot)$, it is feasible to introduce a debiasing term or even incorporate human feedback into the $\Delta(\cdot,\cdot)$ to improve the quality of text generation. We will consider delving deeper into this aspect in future research.

---

> > ### Comment · Reviewer_Mggy · 2023-08-16
> > **The response is read.**
> >
> > Thanks for your response.

---

> > > ### Author Response · Authors · 2023-08-17
> > >
> > > Dear reviewer,
> > >
> > > Thanks for the comment. We wanted to check if we were able to resolve some of your concerns/questions and if you had any further comments on our work. If not, we hope our response may merit raising your score.

---

### Official Review · Reviewer_dohg · 2023-07-06

**Soundness:** 3 good
**Presentation:** 2 fair
**Contribution:** 3 good
**Rating:** 5
**Confidence:** 4

**Summary:**

The traditional approach for memory retrieval is constrained by the quality of the fixed corpus from which memory is retrieved. Based on the exploration that "better generation also promotes better memory", this work proposes the Selfmem framework, which iteratively employs a retrieval-augmented generator to create an unbounded memory pool and uses a memory selector to select a generated memory for the next round. Experimental results demonstrate the effectiveness of Selfmem in fine-tuned small models and few-shot LLMs for three different text generation tasks.

**Strengths:**

1. The motivation of this work is novel. It iteratively employs a retrieval-augmented generator to create an unbounded memory pool and uses a memory selector to select a generated memory for the next round. This enables the model to leverage its own output to improve generation.
2. The method proposed in this paper has a good effect and has achieved improvement on multiple datasets on three text generation tasks. In addition, this work also verified the proposed method for LLM.
3. The experimental analysis of this paper is sufficient. In addition to verifying the effectiveness of the proposed method on multiple datasets, this paper also conducts thorough analyses of each component in the Selfmem to identify bottlenecks and provide insights for future research.


**Weaknesses:**

1. The generality of the method is limited. When the proposed method uses the memory selector to select a candidate from the candidate pool, different metrics are used for different text generation tasks, which limits the generality of the method on different tasks.

2. The cost of the proposed method is large. The Selfmem needs to execute retrieval and generation multiple times until converges.

3. There are some writing problems in this paper:
- There is an extra word “open” in line 215.
- Line 141 \phi(R) -> \phi(\mathcal{R}).
- Equation (4) MemoryEncoder(m).
- No details about how to generate candidate pool \mathbb{C}.

4. A generation example can be provided in the experimental results so that readers can feel the effect of the method in this paper more intuitively.

**Questions:**

1. When the proposed method uses the memory selector to select a candidate, different metrics are used for different text generation tasks, which limits the generality of the method on different tasks. Is there a good general method or metric that can be applied to different text generation tasks? How well does the proposed method perform when using a generic method or metric?

2. How to preserve diversity when generating candidate pool \mathbb{C}?

3. How to draw the conclusion in Line 143 that 'the generator does not need to resort to external memory'?  The conclusion in Line 145 also confuses me a lot, e.g. 'This observation motivates us to select memory according to metrics'. Could you explain in more detail?

4. For tasks such as dialogue generation, this work only uses automatic evaluation, lacking human evaluation. Automated assessments also lack justification.

**Limitations:**

The authors discuss limitations in the appendix.

---

> ### Author Rebuttal · Authors · 2023-08-06
>
> Firstly, we would like to express our sincere gratitude for the time and effort you have dedicated to reviewing our paper. We will address each comment in a point-by-point manner.
>
> - Comment 1: Limited generality and universal metrics for different text generation tasks.
>
> Response 1:
> - We are grateful for the reviewer's insightful comment concerning the generality and applicability of universal metrics across various text generation tasks. While our work indeed employs different metrics for specific generation tasks, we respectfully disagree with the notion that this approach restricts the generality of our framework. Instead, we contend that this demonstrates the versatility and adaptability of our method.
> - The rationale for utilizing distinct metrics for individual tasks stems from the unique characteristics and objectives of each generation task, and what our framework does is to provide a method to optimize these attributes accordingly. For instance, in neural summarization tasks, the ROUGE score is deemed more suitable than the BLEU score for several reasons, including its emphasis on recall over precision of ground truth summaries and the absence of a length penalty for summaries of varying lengths. The ROUGE score has also been shown to achieve stronger correlation with human judgments[1]. This rationale informed our decision to use ROUGE for optimizing our memory selector and, consequently, our generator. If a generation task needs diversity, we optimize for diversity. If it needs fidelity, we optimize for fidelity. The only modification needed is adjusting the memory selection metric in Algorithm 1 of our paper.
> - Furthermore, we believe that the experimental results obtained across three different tasks within this single framework provide empirical evidence that our framework is not a task-specific one.
> - As for the interesting question about the the existence of a universal metric suitable for all text generation tasks, irrespective of their specific requirements, we consider such a metric unlikely. If one were available, there would be no need to divide text generation tasks into multiple subfields. However, one potential solution to this challenge is the use of human preference, as employed in the optimization of Language Models (LLMs) like ChatGPT. Our framework can also be adapted to this context by adjusting the memory selection metric to reflect human preference.
>
> We express our sincere gratitude for raising this thought-provoking issue and welcome further discussion on the subject.
>
>  [1] AlexanderR. Fabbriet al.SummEval: Re-evaluating Summarization Evaluation
>
> ---
>
> - Comment 2: Latency caused by selfmem
> - Response 2: We appreciate the reviewer's concern regarding the latency caused by our method, as mentioned in the limitations section in the Appendix. We acknowledge that our proposed method may result in higher latency compared to conventional generation scenarios. However, it is essential to note that this paper represents the first attempt in this direction, and our primary focus is on the novel aspects of the approach rather than efficiency. We believe there is significant potential for improvement in future work, such as incorporating an efficient dual encoder in place of a cross-encoder for memory selection, and reducing the number of iterations by increasing the size of the candidate pool, as suggested by Reviewer LVYL. We are eager to explore these possibilities and engage in further discussions to address the issue of latency while maintaining the integrity and effectiveness of our approach.
>
> ---
>
> - Comment 3: Writing problems
> - Responses 3: Thank you for your thorough review and we will complete another round of proofreading in our final version. As for the candidate pool generation method, we have included it in Section 4.2: "For all tasks, the candidate generation method involves a beam search with a beam width of 50.”
>
> ---
>
> - Comment 4: Generation example
> - Response 4: Thank you for bringing this to our attention. Upon the paper's acceptance, we will release all outputs, including Selfmem and baselines. Additionally, we will endeavor to find examples that best illustrate our method.
>
> ---
>
> - Comment 5: Preserving diversity in candidate pool
> - Response 5: As demonstrated in previous work [1], employing beam search is adequate for achieving significant improvement, and thus, we have not incorporated diversity into this paper's scope.
>
>     [1] Ann Lee, et al. Discriminative Reranking for Neural Machine Translation
>
> ---
> - Comment 6: Explanation of the conclusion in Line 143 and Line 145
> - Response 6: We apologize for any confusion that our presentation may have caused, and appreciate the opportunity to clarify. The conclusion in Line 143 is drawn from the observation that the generation confidence of the overlapping token set was generally higher than the average. This indicates that the generator is more likely to produce these low-perplexity tokens even without the provided memory. As a result, the conclusion from Line 145 could be more accurately stated as: "This observation motivates us to reconsider selecting memory based on p(y|x)." If there is any remaining confusion, we encourage you to refer to our Response 2 to Reviewer Z8iX.
>
> ---
> - Comment 7: Lack of human evaluation results
> - Response 7: We appreciate your suggestion and have included human evaluation results as well as GPT-4 evaluation results in the attached rebuttal global rebuttal PDF file for your reference.

---

> > ### Comment · Reviewer_dohg · 2023-08-16
> > **For Comment 1**
> >
> > > While our work indeed employs different metrics for specific generation tasks, we respectfully disagree with the notion that this approach restricts the generality of our framework. Instead, we contend that this demonstrates the versatility and adaptability of our method.
> >
> > I do not fully agree with this statement, because if the proposed framework is a better optimizer towards a specific metric, its performance will definitely be better than the other method that does not optimize the whole system using the metric. Additionally, the comparison with the paper 'REPLUG: Retrieval-Augmented Black-Box Language Models' is also needed, because it optimizes IR directly using the final metric.
> >
> > > However, one potential solution to this challenge is the use of human preference, as employed in the optimization of Language Models (LLMs) like ChatGPT. Our framework can also be adapted to this context by adjusting the memory selection metric to reflect human preference.
> >
> > Indeed, your perspective holds merit. Exploring this avenue in the future might yield intriguing results.

---

> > > ### Author Response · Authors · 2023-08-16
> > >
> > > > I do not fully agree with this statement, because if the proposed framework is a better optimizer towards a specific metric, its performance will definitely be better than the other method that does not optimize the whole system using the metric.
> > >
> > > We appreciate your comment and we totally agree with you that systems designed to optimize specific metrics tend to outperform those that don't. Indeed, this is what reward model aims to achieve, as discussed in Section 2.2 (Related Work) of our paper.
> > >
> > > However, the reranking process is just a one-way process, while our model, innovatively utilizes the duality of retrieval-augmented generation and combines the retriever and ranker in a unified framework, which has never been explored before. Furthermore, our experimental results demonstrate that our framework can outperform the two-stage reward model, as evidenced in Table 3 under the "self" column, and in Table 5, where BRIO serves as a reranking model.
> > >
> > > >  Additionally, the comparison with the paper 'REPLUG: Retrieval-Augmented Black-Box Language Models' is also needed, because it optimizes IR directly using the final metric.
> > >
> > > Thanks for your insight question. Here we would like to clarify that the use of the final metric to optimize IR was not first proposed by REPLUG, as we have already included a more relevant baseline, MonoNMT, in our paper. MonoNMT is the pioneering work that employs a learnable IR to enhance retrieval-augmented generation. To our understanding, REPLUG primarily focuses on augmenting **black-box** generation systems with learnable IR, which is not a focusing point of our framework.
> > >
> > > Furthermore, we want to emphasize that both MonoNMT and REPLUG, using learnable retriever to fetch better memory, address the primary problem discussed in our paper: better memory prompts better generation. Our work, on the other hand, is the first to explore the dual problem—better generation also prompts better memory—and combines these two problems. Thank you for providing this REPLUG paper; we will incorporate it into our final version.
> > >
> > > > Indeed, your perspective holds merit. Exploring this avenue in the future might yield intriguing results.
> > >
> > > Thanks for your acknowledgement. We greatly value your expertise and proficiency in retrieval-augmented generation and reward modeling. We hope this additional clarification can address your concerns and merit raising your score. Please kindly let us know if you have any remaining concerns.

---

> > ### Comment · Reviewer_dohg · 2023-08-16
> > **For Comment 2**
> >
> > Thanks to the authors address most of my concerns.
> >
> > > We appreciate the reviewer's concern regarding the latency caused by our method, as mentioned in the limitations section in the Appendix.
> >
> > I find Appendix only mentions this problem while no detail analyzes or further experiments on the latency. An in-depth quantitative analysis is required.

---

> > > ### Author Response · Authors · 2023-08-16
> > >
> > > > I find Appendix only mentions this problem while no detail analyzes or further experiments on the latency. An in-depth quantitative analysis is required.
> > >
> > > Thanks for your comment. We have included in-depth quantitative analysis in the global rebuttal pdf file for your reference.

---

### Official Review · Reviewer_Z8iX · 2023-07-06

**Soundness:** 3 good
**Presentation:** 2 fair
**Contribution:** 2 fair
**Rating:** 5
**Confidence:** 4

**Summary:**

This paper aims to address the retrieval-augmented text generation problem, where the principle of memory retrieval is to select examples similar to the input. The authors are motivated by an observation in their preliminary experiments that the memory examples that better resemble the data distribution during inference come from the model's output, instead of the training data. Therefore, they propose a novel framework, selfmem to address the retrieval-augmented text generation problem. The proposed framework consists of a memory-augmented generator and a memory selector. Differently, the generator can produce multiple candidates to serve as generated memory instead of retrieved memory from the training corpus. The authors experiment on three generation tasks: machine translation, summarization, and dialogue, and demonstrate performance improvement in various evaluation metrics.

**Strengths:**

1. The paper provides sufficient related work, which is helpful for readers to understand the research context.
2. The research motivation is well-supported by preliminary experimental results, which is a good starting point for the research.
3. The paper includes thorough experiments covering representative tasks in the generation field, including machine translation, summarization and dialogue generation, which could be a valuable contribution.

**Weaknesses:**

1. The novelty of the paper is limited. The authors propose a dual structure: Retrieval-augmented Generator and memory selector, each of which is built upon existing paradigms.

2. The overall paper is not well-presented.

(a)  Many parts of the paper lack clear descriptions. For example, the setting of the preliminary experimentation is unclear at all.

In L119-120, how does the number 38.89 in Table 1 come from? For example, what does the head of row (Memory, Hypothesis) mean? There is no brief description of this experimental setting in the context.  The results in Table 1 are not clear enough to understand the motivation.

(b) Additionally, there are grammar errors and unclear sentence structures in the paper that hinder the reader's comprehension.

In L113-114, "The primary motivation behind our framework stems from the observation that the memory, which is more similar in distribution to the data during inference, is not the training data (38.89 BLEU, as shown in Table 1)." -> "The primary motivation behind our framework stems from the observation that the memory is more similar in distribution to the data during inference, not the training data (38.89 BLEU, as shown in Table 1)."

3. The experimental results are not that convincing as the authors did not provide human evaluation results.  Due to the limitations of automatic evaluation metrics, it is usually necessary to conduct human evaluation to assess the quality of generated output in generation tasks.

In this paper, the generation is boosted with the examples that meet the automatic evaluation metrics. This naturally leads to an increase in the numerical values of automatic evaluation metrics.  It is still necessary to use human evaluation results to further determine the quality of the generated output.

**Questions:**

Suggestions:
1. In Figure 2, the shading in the square exceeds the boundaries, which needs to be fixed.
2. More captions are needed for Figure 2(b) to better illustrate the content.

Typos & Grammar errors:
1. Please refer to the comments in Weaknesses.
2. In L215, there is an irrelevant word "open" that should be removed.

**Limitations:**

I didn't find the relevant discussion is this paper.

---

> ### Author Rebuttal · Authors · 2023-08-06
>
> Firstly, we would like to express our sincere gratitude for the time and effort you have dedicated to reviewing our paper. We will address each comment in a point-by-point manner.
>
> - Comment 1: Limited novelty
> - Response 1: We acknowledge that our study's two primary components, namely the retrieval-augmented generator and the two-stage reward model, are indeed rooted in existing literature. We have conscientiously cited these sources throughout our paper to duly credit their contributions. However, we contend that the novelty of our work should not be downplayed, analogous to how GAN (Ian J. Goodfellow et al.) should not be viewed as merely a fusion of a generator and a discriminator. Our research, for the first time, delves into the fundamental limitations of bounded memory within the current retrieval-augmented literature and proposes a novel self-uplifting framework by integrating the primal and dual problems. This unique aspect of our study has also been acknowledged by other Reviewers. Furthermore, we have executed a rigorous empirical evaluation across a diverse range of text generation tasks, encompassing three distinct categories, two generation paradigms, and seven datasets. The results reveal that our method consistently outperforms existing works employing either one of the two components.
>
> ---
>
> - Comment 2: Unclear description of preliminary experiment and grammar errors.
> - Response 2: First, we apologize for any confusion that our presentation may have caused. To clarify, the value 38.89 in Table 1 denotes the average BLEU score of the retrieved memory in relation to the ground-truth reference. In this table, the number under the head(Memory/Hypothesis) refers to the quality of the memory and the hypothesis, as assessed by the BLEU score. The underlying motivation and rationale for this preliminary section can be summarized as follows:
>     1. Given the primal problem that better memory prompts better generation, we first investigate whether the output from the generation model could directly serve as a more effective memory, considering its higher similarity to the ground truth compared to the retrieved memory (please refer to the first and second rows under the Memory head in Table 1).
>     2. The results do not support this assumption (as seen in the first and second rows under the Hypothesis head in Table 1), prompting us to consider two potential reasons for this outcome. We then conduct another experiment to eliminate the first possibility (evident in the third row of Table 1) and opt for the second reason, supported by uncertainty analysis. This finding encourages us to avoid selecting memory based on p(y|x) and reaffirms the necessity of a memory selector in our framework.
>
>     We hope that this explanation clarifies our motivation and we are more than willing to engage in further discussions to address any potential misunderstandings.
>
> ---
>
> - Comment 3: Lack of human evaluation results
> - Response 3: We appreciate your suggestion and have included human evaluation results as well as GPT-4 evaluation results in the attached global rebuttal PDF file for your reference.
>
> ---
>
> - Comment 4: Lack of discussion of limitation
> - Response 4: The discussion can be found in the appendix, which is included in the supplementary zip file.

---

> ### Author Response · Authors · 2023-08-18
>
> Dear Reviewer, I hope this message finds you well. As the discussion period for our paper  is nearing its end, we kindly request your response to our rebuttal letter. We value your insights and are eager to address any concerns you may have. Your timely feedback would be greatly appreciated and will help us improve our paper. Thank you for your time and dedication to the review process.

---

### Official Review · Reviewer_LVYL · 2023-07-07

**Soundness:** 3 good
**Presentation:** 2 fair
**Contribution:** 3 good
**Rating:** 6
**Confidence:** 3

**Summary:**

The paper proposes an iterative text generation procedure. First, the authors augment the model with retrieval to generate the initial beam of predictions. Then they apply rescoring model (trained separately to maximize BLEU/ROUGE scores) to select the higher-quality predictions. Finally, the authors augment the model with these predictions and perform the next round of generation. The procedure is then repeated multiple times.

**Strengths:**

The paper performs experiments on multiple benchmarks for machine translation and summarization, showing a strong performance of the method. The authors also experiment with several models, including (1) fine-tuned fusion-in-decoder, (2) fine-tuned fusion-in-decoder where input and retrievals encoded separately, (3) frozen decoder-only LLMs

**Weaknesses:**

The paper could be restructured to make it easier to read. Specifically, the paper heavily relies on the notion of "memory." This could be confusing since the paper only briefly explains what specific memory the method is using. For example, do we retrieve parallel sentences from the training data in machine translation? or do we retrieve from the monolingual corpus in source / target languages?

Moreover, I'm not sure I'm a fan of the "memory" positioning of the paper. Does it matter that the first prediction is made using retrieval augmented model?

Finally, it's unclear to me how much gains are due to the reward model and how much gains are due to iterative prediction. For example, while we know that doing four iterations works better than 1 -- would you similar gains from doing 1 iteration but with four times as large beam size? This way, the reward model could pick higher-quality predictions.

**Questions:**

Would it make sense to cite related works on the iterative prediction? For example, some of the potential candidates include
Zemlyanskiy et al., Generate-and-Retrieve: use your predictions to improve retrieval for semantic parsing, COLING 2022
Kumar et al., In-context Examples Selection for Machine Translation, arxiv

What is the "Memory" column in Tables 1 and 3? Does it mean the highest (or average?) BLEU score of the retrieved sample if we use it as a prediction? If that's so, then I'm surprised that such a nearest neighbord -like classifier gets > 38.89 BLEU (Table 1).

**Limitations:**

Yes

---

> ### Author Rebuttal · Authors · 2023-08-06
>
> Firstly, we would like to express our sincere gratitude for the time and effort you have dedicated to reviewing our paper. We will address each comment in a point-by-point manner.
> - Comment 1: Regarding the explanation and positioning of memory
> - Response 1: We maintain that memory plays a crucial role in our system and deserves emphasis. To illustrate this, let us consider a machine translation task as an example. In the first step, we perform a retrieval operation in bilingual corpora, using the retrieved sentence (target side) as our initial memory, as outlined in line 1 of Algorithm 1 and lines 154-155 of the paper. Subsequently, we employ the retrieved memory to train a memory-augmented generator, enabling us to leverage the selected memory in the subsequent iterative process. While it is not essential for the first prediction to be made using a retrieval-augmented model, it is indeed crucial for us to have a retrieval-augmented model to utilize the first prediction (after selection by the memory selector).
>
> ---
> - Comment 2: Would you achieve similar gains from conducting one iteration but with a beam size four times larger?
> - Response 2: We acknowledge the possibility of further improvement by increasing the candidate pool size within a single iteration round. However, we believe that such a setting may not be highly relevant to the main contribution of our paper. As demonstrated in Figure 2 of [1] and Figure 2 of [2], it is a well-established conclusion that a larger candidate pool results in higher performance for a reward model. Our paper distinctly demonstrates that the iterative process within the Selfmem framework can also yield superior outcomes. As illustrated in Figure 3(b) of our paper, the quality of the candidate set (measured in terms of oracle, quartile, average, and minimum scores) consistently improves throughout the iteration process. This result could not be attained solely by increasing the candidate pool size. The core essence of the framework lies in the steady improvement between the generator and reward model. We argue that an expanded candidate set can be viewed as an enhancement of our framework without changing its fundamental nature. Consequently, the potential outcome of an enlarged candidate pool would manifest as an overall upward shift in Figure 3(b).
>
> ---
>
> - Comment 3: Missing reference
> - Response 3: We appreciate your attention to this matter. We concur that the two references you mentioned are indeed relevant to our paper, particularly the first one by Zemlyanskiy et al. We will ensure their inclusion in the final version of our paper.
>
> ---
>
> - Comment 4: Memory column in Table 1 and 3
> - Response 4: The memory columns featured in Tables 1 and 3 display the average BLEU scores calculated from the retrieved samples. To mitigate any potential confusion, we will enhance these tables with more detailed captions. The high BLEU score achieved by a nearest-neighbor-like classifier can primarily be attributed to the dataset we have utilized. Our choice is the JRC-Acquis dataset, which consists of parallel legislative texts pertaining to European Union law and applicable to its member states. Owing to its highly relevant and well-structured data, this corpus serves as an exemplary testbed for assessing memory-augmented neural machine translation (NMT) systems. As a result, a significant number of pertinent studies[3][4] make use of this dataset, which is why we have also opted for it.
>
> [1] Ann Lee, et al. Discriminative Reranking for Neural Machine Translation
>
> [2] Liu, Yixin, et al. SimCLS: A simple framework for contrastive learning of abstractive summarization
>
> [3] Gu, Jiatao, et al. Search engine guided neural machine translation
>
> [4] Cai, Deng, et al. Neural machine translation with monolingual translation memory

---

> ### Author Response · Authors · 2023-08-18
>
> Dear Reviewer, I hope this message finds you well. As the discussion period for our paper  is nearing its end, we kindly request your response to our rebuttal letter. We value your insights and are eager to address any concerns you may have. Your timely feedback would be greatly appreciated and will help us improve our paper. Thank you for your time and dedication to the review process.

---

### Author Rebuttal · Authors · 2023-08-07

Dear Reviewers,

We would like to express our deepest appreciation for the time and effort you have devoted to reviewing our conference paper. In response to the questions raised by each reviewer, we have provided detailed answers in the corresponding sections, and we hope that these clarifications will address some of your concerns.

In light of the shared concerns regarding latency evaluation and human evaluation, we have prepared an additional PDF file that is attached to this response for your reference. We are eager to engage in further discussions about our paper and to continue refining and enhancing our work with the invaluable assistance of esteemed reviewers.

Once again, thank you for your valuable insights, and we look forward to your continued guidance and support.

Sincerely,

Authors

---

### Author Response · Authors · 2023-08-15

Dear Reviewers,

We wanted to check if we were able to resolve some of your concerns/questions and if you had any further comments on our work. We will be happy to address any additional concerns that you might have, and we look forward to engaging in further discussions on our paper.

Thanks so much for the time and effort you have devoted to our paper.

---

### Decision · Program_Chairs · 2023-09-21

**Decision:**

Accept (poster)

**Comment:**

The paper empirically demonstrates that an intuitive iterative candidate generation scheme can lead to a well-performing model for retrieval-augmented text generation. The idea is sensible and a nice extension of recent generate-rather-than-retrieve line of works (Yu et al., 2023; inter alia), and the performance is strong. However, the paper makes a "memory"-based narrative which makes the technical content obfuscated, and at the same time leaves a few experimental settings underspecified (but the code is supplied). Despite the subpar presentation, the technical contributions seem strong enough to justify an acceptance.